# ARE SPECTRAL AUGMENTATIONS NECESSARY IN CONTRAST-BASED GRAPH SELF-SUPERVISED LEARNING?

## ABSTRACT

The recent surge in contrast-based graph self-supervised learning has prominently featured an intensified exploration of spectral cues. Spectral augmentation, which involves modifying a graph's spectral properties such as eigenvalues or eigenvectors, is widely believed to enhance model performance. However, an intriguing paradox emerges, as methods grounded in seemingly conflicting assumptions or heuristic approaches regarding the spectral domain demonstrate notable enhancements in learning performance. This paradox raises the critical question of whether spectral augmentations are really necessary for contrast-based graph self-supervised learning. This study undertakes an extensive investigation into this inquiry, conducting a thorough study of the relationship between spectral characteristics and the learning outcomes of contemporary methodologies. Based on this analysis, we claim that the effectiveness and significance of spectral augmentations need to be questioned. Instead, we revisit simple edge perturbation: random edge dropping designed for node-level self-supervised learning and random edge adding intended for graph-level self-supervised learning. Compelling evidence is presented that these simple yet effective strategies consistently yield superior performance while demanding significantly fewer computational resources compared to existing spectral augmentation methods. The proposed insights represent a significant leap forward in the field, potentially reshaping the understanding and implementation of graph self-supervised learning.

## 1 INTRODUCTION

In recent years, graph learning has emerged as a powerhouse for handling complex data relationships in multiple fields, offering vast potential and value, particularly in domains such as data mining (Hamilton et al., 2017), computer vision (Xu et al., 2017), network analysis (Chen et al., 2020b), and bioinformatics (Jin et al., 2018). However, limited labels make graph learning challenging to apply in real-world scenarios. Inspired by the great success of Self-Supervised Learning (SSL) in other domains (Devlin et al., 2018; Chen et al., 2020a), Graph Self-Supervised Learning (Graph SSL) has made rapid progress and has shown promise by achieving state-of-the-art performance on many tasks (Xie et al., 2022), where **C**ontrast-based **G**raph **SSL** (**CG-SSL**) are most dominant (Liu et al., 2023). This type of method is grounded in the concept of mutual information (MI) maximization. The primary goal is to maximize the estimated MI between augmented instances of the same object, such as nodes, subgraphs, or entire graphs. Among the new developments in **CG-SSL**, approaches inspired by graph spectral methods have garnered significant attention. A prevalent conviction is that spectral information, including the eigenvalues and eigenvectors of the graph's Laplacian, plays a crucial role in enhancing the efficacy of **CG-SSL** (Liu et al., 2022a; Ko et al., 2023; Lin et al., 2023; Yang et al., 2023; Chen et al., 2024).

In general, methods in **CG-SSL** can be categorized into two types based on whether augmentation is performed on the input graph to generate different views (Chen et al., 2024). i.e. augmentation-based and augmentation-free methods. Of the two, the augmentation-based methods are more predominant and widely studied (Hassani & Khasahmadi, 2020; Liu et al., 2023; You et al., 2020; Liu et al., 2022a; Lin et al., 2023; Yang et al., 2023). Specifically, spectral augmentation has received significant attention, as it modifies a graph's spectral properties. This approach is believed to enhance model

performance, aligning with the proposed importance of spectral information in **CG-SSL**. However, there seems no consensus on the true effectiveness of spectral information in the previous works proposing and studying spectral augmentation. SpCo (Liu et al., 2022a) introduces the general graph augmentation (GAME) rule, which suggests that the difference in high-frequency parts between augmented graphs should be larger than that of low-frequency parts. SPAN (Lin et al., 2023) contends that effective topology augmentation should prioritize perturbing sensitive edges that have a substantial impact on the graph spectrum. Therefore, a principled augmentation method is designed by directly maximizing spectral change with a certain perturbation budget, without mentioning any specific domain of spectrum. GASSER (Yang et al., 2023) selectively perturbs graph structures based on spectral cues to better maintain the required invariance for contrastive learning frameworks. Specifically, it aims to augment the graphs to preserve task-relevant frequency components and perturb the task-irrelevant ones with care. While all three related methods are augmentation-based and share in the set of **CG-SSL** frameworks like GRACE (Zhu et al., 2020) and MVGRL (Hassani & Khasahmadi, 2020), a contradiction emerges among these related works on spectral augmentation: while SPAN advocates for **maximizing the distance** between the spectrum of augmented graphs regardless of spectral domains, SpCo and GASSER argue for the **preservation** of specific spectral components and domains during augmentation. The consistent performance gain derived from opposing methodical designs naturally raises our concern:

- *Are spectral augmentations necessary in contrast-based graph SSL?*

Given the question, this study aims to critically evaluate the effectiveness and significance of spectral augmentation in contrast-based graph SSL frameworks (**CG-SSL**). With evidence-supported claims and findings in the following sections, we can give a negative answer to the question above: **No, they are not very effective and we don't really need them.** To be specific, we find that spectral augmentation does not significantly contribute to the learning efficacy while more straightforward edge perturbations are already good enough for **CG-SSL**. We manage to elaborate on our conclusion through a series of studies carried out in the following efforts:

1. In Sec. 4, we explore the dependency of spectral augmentation effectiveness on the depth of the network, positing that shallower networks with fewer convolutional layers perform better but demonstrate diminished benefits from spectral changes.

2. In Sec 5 We claim that simple edge perturbation techniques, like adding edges to or dropping edges from the graph, not only compete well but often outperform spectral augmentations, without any significant help from spectral cues. To support this,

   **(a)** In Sec. 6, overall model performance on test accuracy with four state-of-the-art frameworks on both node- and graph-level classification tasks support the superiority of simple edge perturbation. **(b)** Studies in Sec. 7.1 reveal the indistinguishability between the average spectrum of augmented graphs from edge perturbation with optimal parameters on different datasets, no matter how different that of original graphs is, indicating GNN encoders can hardly learn spectral information from augmented graphs. That is to say, edge perturbations can not benefit from spectral information. **(c)** In Sec. 7.2, we analyze the effectiveness of state-of-the-art spectral augmentation baseline (*i.e.*, SPAN) by perturbing edges to alter the spectral characteristics of augmented graphs from simple edge perturbation augmentation and examining the impact on model performance. As it turns out, the results show no performance degradation, indicating the spectral information contained in the augmentation is not significant to the model performance. **(d)** In Appendix E.4, statistical analysis is carried out to argue that the major reason edge perturbation works well is not because of the spectral information as they are statistically not the key factor on model performance.

## 2 RELATED WORK

**Contrast-based Graph Self-Supervised (CG-SSL).** **CG-SSL** learning alleviates the limitations of supervised learning, which heavily depends on labeled data and often suffers from limited generalization (Liu et al., 2022b). This makes it a promising approach for real-world applications where labeled data is scarce. **CG-SSL** applies a variety of augmentations to the training graph to obtain augmented views. These augmented views, which are derived from the same original graph, are treated as positive sample pairs or sets. The key objective of **CG-SSL** is to maximize the mutual information between

these views to learn robust and invariant representations. However, directly computing the mutual information of graph representations is challenging. Hence, in practice, **CG-SSL** frameworks aim to maximize the lower bound of mutual information using different estimators such as InfoNCE (Gutmann & Hyvärinen, 2010), Jensen-Shannon (Nowozin et al., 2016), and Donsker-Varadhan (Belghazi et al., 2018). For instance, frameworks like GRACE (Zhu et al., 2020), GCC (Qiu et al., 2020), and GCA (Zhu et al., 2021b) utilize the InfoNCE estimator as their objective function. On the other hand, MVGRL (Hassani & Khasahmadi, 2020) and InfoGraph (Sun et al., 2019) adopt the Jensen-Shannon estimator. Some **CG-SSL** methods explore alternative principles. G-BT (Bielak et al., 2022) extends the redundancy-reduction principle, minimizing dissimilarity between metrics from two augmented graph views. BGRL (Thakoor et al., 2021) adopts a momentum-driven Siamese architecture, using node feature masking and edge modification as augmentations to maximize mutual information between online and target network representations.

**Graph Augmentations in CG-SSL.** Beyond the choice of objective functions, another crucial aspect of augmentation-based methods in **CG-SSL** is the selection of augmentation techniques. Early work by (Zhu et al., 2020) and (You et al., 2020) introduced several domain-agnostic heuristic graph augmentation for **CG-SSL**, such as edge perturbation, attribute masking, and subgraph sampling. These straightforward and effective methods have been widely adopted in subsequent **CG-SSL** frameworks due to their demonstrated success (Thakoor et al., 2021; Yu et al., 2024). However, these domain-agnostic graph augmentations often lack interpretability, making it difficult to understand the exact impact of these augmentations on the graph structure and learning outcomes. To address this issue, MVGRL (Hassani & Khasahmadi, 2020) introduces graph diffusion as an augmentation strategy, where the original graph provides local structural information and the diffused graph offers global context. MVGRL demonstrates experimentally that by optimizing for consistency between node representations from these two perspectives, it's possible to obtain representations that encode both local and global structural information. Moreover, three spectral augmentation methods–SpCo (Liu et al., 2022a), GASSER (Yang et al., 2023), and SPAN (Lin et al., 2023)–stand out by offering design principles based on spectral graph theory, focusing on how to enhance **CG-SSL** performance through spectral manipulations. However, our explorations show that these methods are unable to consistently outperform heuristic graph augmentations such as edge perturbation (DROPEDGE or ADDEDGE) in terms of performance under fair comparisons, and thus the design principles of graph augmentation still require further validation.

## 3 PRELIMINARY STUDY

**Contrast-based graph self-supervised learning framework. CG-SSL** captures invariant features of a graph by generating multiple views (typically two) through augmentations and then maximizing the mutual information between these views (Xie et al., 2022). This approach is ultimately used to improve performance on various downstream tasks. Following previous work (Wu et al., 2021; Liu et al., 2022b; Xie et al., 2022), we first denote the generic form of the augmentation $\mathcal{T}$ and objective functions $\mathcal{L}_{cl}$ of graph contrastive learning. Given a graph $\mathcal{G} = (\mathbf{A}, \mathbf{X})$ with adjacency matrix $\mathbf{A}$ and feature matrix $\mathbf{X}$, the augmentation is defined as the transformation function $\mathcal{T}$. In this paper, we are mainly concerned with topological augmentation, in which feature matrix $\mathbf{X}$ remains intact:

$$\widetilde{\mathbf{A}}, \widetilde{\mathbf{X}} = \mathcal{T}(\mathbf{A}, \mathbf{X}) = \mathcal{T}(\mathbf{A}), \mathbf{X} \tag{1}$$

In practice, two augmented views of the graph are generated, denoted as $\mathcal{G}^{(1)} = \mathcal{G}(\mathcal{T}_1(\mathbf{A}, \mathbf{X}))$ and $\mathcal{G}^{(2)} = \mathcal{G}(\mathcal{T}_2(\mathbf{A}, \mathbf{X}))$. The objective of GCL is to learn representations by minimizing the contrastive loss $\mathcal{L}_{cl}$ between the augmented views:

$$\theta^*, \phi^* = \arg\min_{\theta, \phi} \mathcal{L}_{cl} \left( p_\phi \left( f_\theta \left( \mathcal{G}^{(1)} \right), f_\theta \left( \mathcal{G}^{(2)} \right) \right) \right), \tag{2}$$

where $f_\theta$ represents the graph encoder parameterized by $\theta$, and $p_\phi$ is a projection head parameterized by $\phi$. The goal is to find the optimal parameters $\theta^*$ and $\phi^*$ that minimize the contrastive loss.

In this paper, we utilize four prominent **CG-SSL** frameworks to study the effect of spectral: MVGRL, GRACE, BGRL, and G-BT. MVGRL introduces graph diffusion as augmentation, while the other three frameworks use edge perturbation as augmentation. Each framework employs different strategies for its contrastive loss functions. MVGRL and GRACE use the Jensen-Shannon and InfoNCE

estimators as object functions, respectively. In contrast, BGRL and G-BT adopt the BYOL loss (Grill et al., 2020) and Barlow Twins loss (Zbontar et al., 2021), which are designed to maximize the agreement between the augmented views without relying on negative samples. A more detailed description of the loss function can be found in the Appendix C.

**Graph spectrum & Definition and application of spectral augmentation.** We follow the standard definition of graph spectrum in this study, details of which can be found in Appendix B. Among various augmentation strategies proposed to enhance the robustness and generalization of graph neural networks, spectral augmentation has been considered a promising avenue (Lin et al., 2023; Liu et al., 2022a; Bo et al., 2023; Yang et al., 2023). Spectral augmentation typically involves implicit modifications to the eigenvalues of the graph Laplacian, aiming at enhancing model performance by encouraging invariance to certain spectral properties. Among them, SPAN achieved state-of-the-art performance in both node classification and graph classification. In short, SPAN elaborates two augmentation functions, $\mathcal{T}_1$ and $\mathcal{T}_2$, where $\mathcal{T}_1$ maximizes the spectral norm in one view, and $\mathcal{T}_2$ minimizes it in the other view. Subsequently, these two augmentations are implemented in the four **CG-SSL** frameworks mentioned above (Strict definition in Appendix B). The paradigm used by SPAN aims to allow the GNN encoder to focus on robust spectral components and ignore the sensitive edges that can change the spectral drastically when perturbed.

## 4 LIMITATIONS OF SPECTRAL AUGMENTATIONS

**Limitations of shallow GNN encoders in capturing spectral information.** Multiple previous studies indicate that shallow, rather than deep, GNN encoders can be effective in graph self-supervised learning. This might be the result of overfitting commonly witnessed in standard GNN tasks. We have also carried out many empirical studies with a range of **CG-SSL** frameworks and augmentations to support this idea in contrast-based graph SSL. As the most commonly applied GNN encoder in CG-SSL (You et al., 2020; Yu et al., 2024; Guo et al., 2024; Lin et al., 2024), an empirical study on the relationship between the depth of GCN encoder and learning performance is conducted and results are presented in Fig. 1. From that, we can conclude that shallow GCN encoders with 1 or 2 layers usually have the best performance. Note that this tendency is not very clear on graph-level tasks because the embedding of the graph from all layers will be concatenated together to perform prediction. It indicates that a deep encoder has theoretically better expressive power than shallower encoders. Therefore, still better performance of GCN encoders with 1 or 2 layers implies that any more layers are unnecessary and might hurt the quality of the learned representation of the graph.

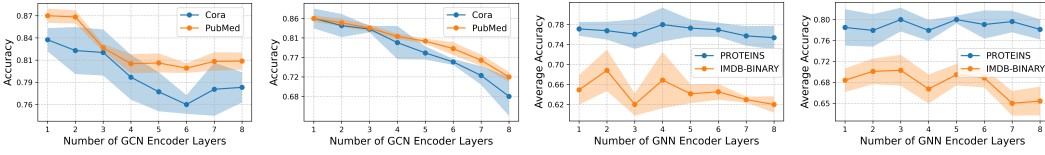

|   (a) G-BT on node CLS | (b) MVGRL on node CLS | (c) G-BT on graph CLS | (d) MVGRL on graph CLS |

Figure 1: Accuracy of **CG-SSL** v.s. number of GCN layers on node and graph classification on representative datasets. (a) G-BT on node classification. (b) MVGRL on node classification. (c) G-BT on graph classification. (d) MVGRL on graph classification. We choose two representative datasets for each task, i.e. CORA and CITESEER for the node-level and PROTEINS and IMDB-BINARY for the graph-level classification. Definition of the accuracy of **CG-SSL**, details of datasets and other experimental settings are mentioned in Section 6.1.

By design, most GNN encoders primarily aggregate local neighborhood information through their layered structure, where each layer extends the receptive field by one hop. The depth of a GNN critically determines its ability to integrate information from various parts of the graph. With only a limited number of layers, a GNN's receptive field is restricted to immediate neighborhoods (e.g., 1-hop or 2-hop distances). This limitation severely constrains the network's ability to assimilate and leverage broader graph topologies or global features that are essential for encoding the spectral properties of the graph, given the definition of the graph spectrum.

**Limited implications for spectral augmentation in CG-SSL.** Given the limitations of shallow GNNs in capturing spectral information, the utility of spectral augmentation techniques in graph

self-supervised learning settings warrants scrutiny. Spectral augmentation involves modifying the spectral components (e.g., eigenvalues and eigenvectors) of a graph to enrich the training process or to create diverse samples for enhancing learning robustness. However, if the primary encoder's architecture—specifically, a shallow GNN—is intrinsically limited in its ability to perceive and process spectral properties, then the benefits of such augmentations are likely to be minimal.

**Gap between spectral theory and graph learning.** Furthermore, beyond the limitations of shallow GNNs in capturing spectral information, there is a significant gap between the theoretical foundations of spectral methods and their practical application in graph learning. These methods often rely on simplifying assumptions that may not hold in real-world scenarios (Liu et al., 2022a; Yang et al., 2023). A detailed discussion of these challenges is provided in Appendix D.

## 5 EDGE PERTURBATION IS ALL YOU NEED

So far, our findings indicate that spectral augmentation is not particularly effective in contrast-based graph self-supervised learning. It may suggest that spectral augmentation essentially amounts to random topology perturbation, based on inconsistencies in previous studies (Lin et al., 2023; Liu et al., 2022a; Yang et al., 2023) and the theoretical insight that a shallow encoder can hardly capture spectral properties. In fact, most of the spectral augmentations are basically performing edge perturbations on the graph with some targeted directions. Since we preliminarily conclude that it is quite difficult for those augmentations to benefit from the spectral properties of graphs, it is very intuitive to hypothesize that edge perturbation itself matters in the learning process.

Consequently, we are turning back to **E**dge **P**erturbation (**EP**), a more straightforward and proven method for augmenting graph data. The two primary methods of edge perturbation are DROPEDGE and ADDEDGE. We want to claim that edge perturbation has a better performance than spectral augmentations and prove empirically that none of them actually or even can benefit much from any spectral information and properties. Also, we demonstrate edge perturbation is much more efficient in practical applications for both time and space sake, where spectral operations are almost infeasible. Overall, we will support the idea with evidence in the following sections that simple edge perturbation is not only good enough but even very optimal in **CG-SSL** compared to spectral augmentations.

Edge perturbation involves modifying the topology of the graph by either removing or adding edges at random. We detail the two main types of edge perturbation techniques used in our frameworks: edge dropping and edge adding.

**DROPEDGE.** Edge dropping is the process of randomly removing a subset of edges from the original graph to create an augmented view. Adopting the definition from (Rong et al., 2020), let $\mathcal{G} = (\mathbf{A}, \mathbf{X})$ be the original graph with adjacency matrix $\mathbf{A}$. We introduce a mask matrix $\mathbf{M}$ of the same dimensions as $\mathbf{A}$, where each entry $M_{ij}$ follows a Bernoulli distribution with parameter $1 - p$ (denoted as the drop rate). The edge-dropped graph $G'$ is then obtained by element-wise multiplication of $\mathbf{A}$ with $\mathbf{M}$ (where $\odot$ denotes the Hadamard product):

$$\mathbf{A}' = \mathbf{A} \odot \mathbf{M} \tag{3}$$

**ADDEDGE.** Edge adding involves randomly adding a subset of new edges to the original graph to create an augmented view. Let $\mathbf{N}$ be an adding matrix of the same dimensions as $\mathbf{A}$, where each entry $N_{ij}$ follows a Bernoulli distribution with parameter $q$ (denoted as the add rate), and $N_{ij} = 0$ for all existing edges in $\mathbf{A}$. The edge-added graph $\mathcal{G}''$ is obtained by adding $\mathbf{N}$ to $\mathbf{A}$:

$$\mathbf{A}'' = \mathbf{A} + \mathbf{N} \tag{4}$$

These two operations ensure that the augmented views $\mathcal{G}^{(1)}$ and $\mathcal{G}^{(2)}$ have modified adjacency matrices $\mathbf{A}'$ and $\mathbf{A}''$ respectively, which are used to generate contrastive views while preserving the feature matrix $\mathbf{X}$.

### 5.1 ADVANTAGE OF EDGE PERTURBATION OVER SPECTRAL AUGMENTATIONS

Edge perturbation offers several key advantages over spectral augmentation, making it a more effective and practical choice for **CG-SSL**. Compared to augmentations related to the graph spectrum, it has three major advantages.

**Theoretically intuitive.** Edge perturbation is inherently simpler and more intuitive. It directly modifies the graph's structure by adding or removing edges, which aligns well with the shallow GNN encoders' strength in capturing local neighborhood information. Given that shallow GNNs have a limited receptive field, they are better suited to leveraging the local structural changes introduced by edge perturbation rather than the global changes implied by spectral augmentation.

**Significantly better efficiency.** Edge perturbation methods such as edge dropping (DROPEDGE) and edge adding (ADDEDGE) are computationally efficient. Unlike spectral augmentation, which requires costly eigenvalue and eigenvector computations, edge perturbation can be implemented with basic graph operations. This efficiency translates to faster training and inference times, making it more suitable for large-scale graph datasets and real-time applications. As shown in Table 1, the time and space complexity of spectrum-related calculations are several orders of magnitude higher than those of simple edge perturbation operations. This makes spectrum-related calculations impractical for the large datasets typically encountered in real-world applications.

Table 1: Time and space complexity of different methods (Empirical Time is on PUBMED dataset)

| Method | Time Complexity | Space Complexity | Empirical Time (s/epoch) |
|---|---|---|---|
| Spectrum calculation | $O(n^3)$ | $O(n^2)$ | 26.435 |
| DROPEDGE | $O(m)$ | $O(m)$ | 0.140 |
| ADDEDGE | $O(m)$ | $O(m)$ | 0.159 |

**Optimal learning performance.** Most importantly and directly, our comprehensive empirical studies indicate that edge perturbation methods lead to significant improvements in model performance, as presented and analyzed in Sec. 6. From the results there, the conclusion can be drawn that the performance of the proposed augmentations is not only better than those of spectral augmentations but also matches or even surpasses the performance of other strong benchmarks.

These advantages position edge perturbation as a robust and efficient method for graph augmentation in self-supervised learning. In the following section, we will present our experimental analysis, demonstrating the accuracy gains achieved through edge perturbation methods.

# 6 EXPERIMENTS ON SSL PERFORMANCE

## 6.1 EXPERIMENTAL SETTINGS

**Task and Datasets.** We conducted extensive experiments for node-level classification on seven datasets: CORA, CITESEER, PUBMED (Kipf & Welling, 2016), PHOTO, COMPUTERS (Shchur et al., 2018), COAUTHOR-CS, and COAUTHOR-PHY. These datasets include various types of graphs, such as citation networks, co-purchase networks, and co-authored networks. Note that we do not include huge-scale datasets like OGBN (Hu et al., 2021) for the high complexity of spectral augmentations. While both DROPEDGE and ADDEDGE have linear complexity that can easily run on those huge datasets, no spectral augmentation can scale to them. Additionally, we carried out graph-level classification on five datasets from the TUDataset collection (Morris et al., 2020), which include biochemical molecules and social networks. More details of these datasets be found in Appendix A.

**Baselines.** We conducted experiments under four **CG-SSL** frameworks: MVGRL, GRACE, G-BT, and BGRL (mentioned in Sec 3), using DROPEDGE, ADDEDGE, and SPAN (Lin et al., 2023) as augmentation strategies. Note that there are only three very relevant studies on spectral augmentation strategies of **CG-SSL** to the authors' best knowledge, i.e., SPAN, SpCo (Liu et al., 2022a) and GASSER (Yang et al., 2023). Among them, GASSER does not have open-sourced code so we are not able to reproduce any related results, but we try our best to directly adopt the best performance reported in that study to ensure comparison is possible. Also, SpCo is only applicable to node-level tasks and its implementation is not robust enough to generalize to all the node-level datasets and **CG-SSL** frameworks. Therefore, we manage to include the results of all the settings that it is feasible to do, which is its original setting and the combination of GRACE and it. Given the infeasibility and inaccessibility of the two, we selected SPAN as a major baseline because it is robust and general enough to all the datasets and experimental settings while allowing the modular plug-and-play integration of edge perturbation methods, enabling a very direct angle to evaluate the effectiveness

Table 2: Node classification. Results of baselines with '†' are adopted directly from previous works. MVGRL+PPR is the original setting of MVGRL. The best results in each cell are highlighted by grey . The best results overall are highlighted with **bold and underline**. Metric is accuracy (%).

| Model | CORA | CITESEER | PUBMED | PHOTO | COMPUTERS | COAUTHOR-CS | COAUTHOR-PHY |
|---|---|---|---|---|---|---|---|
| GCA† | 83.67 ± 0.44 | 71.48 ± 0.26 | 78.87 ± 0.49 | 92.53 ± 0.16 | 88.94 ± 0.15 | 93.10 ± 0.01 | — |
| GMI† | 83.02 ± 0.33 | 72.45 ± 0.12 | 79.94 ± 0.25 | 90.68 ± 0.17 | 82.21 ± 0.31 | 91.08 ± 0.56 | — |
| DGI† | 82.34 ± 0.64 | 71.85 ± 0.74 | 76.82 ± 0.61 | 91.61 ± 0.22 | 83.95 ± 0.47 | 92.15 ± 0.63 | — |
| SpCo | 83.78 ± 0.70 | 71.82 ± 1.26 | 80.86 ± 0.43 | — | — | — | — |
| GASSER† | 85.27 ± 0.10 | 75.41 ± 0.84 | 83.00 ± 0.61 | 93.17 ± 0.31 | 88.67 ± 0.15 | — | — |
| MVGRL + PPR | 83.53 ± 1.19 | 71.56 ± 1.89 | 84.13 ± 0.26 | 88.47 ± 1.02 | 89.84 ± 0.12 | 90.57 ± 0.61 | OOM |
| MVGRL + DROPEDGE | 84.31 ± 1.95 | 74.85 ± 0.73 | 85.62 ± 0.45 | 89.28 ± 0.95 | **90.43 ± 0.33** | 93.20 ± 0.81 | 95.70 ± 0.28 |
| MVGRL + ADDEDGE | 83.21 ± 1.65 | 73.65 ± 1.60 | 84.86 ± 1.19 | 87.15 ± 1.36 | 87.59 ± 0.53 | 92.91 ± 0.65 | 95.33 ± 0.23 |
| MVGRL +SPAN | 84.57 ± 0.22 | 73.65 ± 1.29 | 85.21 ± 0.81 | 92.33 ± 0.99 | 88.75 ± 0.20 | 92.25 ± 0.76 | OOM |
| MVGRL + GASSER† | 80.36 ± 0.05 | 74.48 ± 0.73 | 80.80 ± 0.19 | — | — | — | — |
| G-BT + DROPEDGE | **86.51 ± 2.04** | 72.95 ± 2.46 | 87.10 ± 1.21 | 93.55 ± 0.60 | 88.66 ± 0.46 | **93.31 ± 0.05** | **96.06 ± 0.24** |
| G-BT + ADDEDGE | 82.10 ± 1.48 | 66.36 ± 4.25 | 85.98 ± 0.84 | 93.68 ± 0.79 | 87.81 ± 0.79 | 91.98 ± 0.66 | 95.51 ± 0.02 |
| G-BT + SPAN | 84.06 ± 2.85 | 67.46 ± 3.18 | 85.97 ± 0.41 | 91.85 ± 0.22 | 88.73 ± 0.62 | 92.63 ± 0.07 | OOM |
| GRACE + DROPEDGE | 84.19 ± 2.07 | **75.44 ± 0.32** | **87.84 ± 0.37** | 92.62 ± 0.73 | 86.67 ± 0.61 | 93.15 ± 0.23 | OOM |
| GRACE + ADDEDGE | 85.78 ± 0.62 | 71.65 ± 1.63 | 85.25 ± 0.47 | 89.93 ± 0.74 | 76.74 ± 0.57 | 92.46 ± 0.25 | OOM |
| GRACE + SPAN | 82.84 ± 0.91 | 67.76 ± 0.21 | 85.11 ± 0.71 | **93.72 ± 0.21** | 88.71 ± 0.06 | 91.72 ± 1.75 | OOM |
| GRACE + GASSER† | 84.10 ± 0.26 | 74.47 ± 0.64 | 83.97 ± 0.52 | — | — | — | — |
| GRACE + SpCo | 81.61 ± 0.75 | 70.83 ± 1.47 | 84.97 ± 1.13 | — | — | — | — |
| BGRL + DROPEDGE | 83.21 ± 3.29 | 71.46 ± 0.56 | 86.28 ± 0.13 | 92.90 ± 0.69 | 88.68 ± 0.65 | 91.58 ± 0.18 | 95.29 ± 0.19 |
| BGRL + ADDEDGE | 81.49 ± 1.21 | 69.66 ± 1.34 | 84.54 ± 0.22 | 91.85 ± 0.75 | 86.75 ± 1.15 | 91.78 ± 0.77 | 95.29 ± 0.09 |
| BGRL + SPAN | 83.33 ± 0.45 | 66.26 ± 0.92 | 85.97 ± 0.41 | 91.72 ± 1.75 | 88.61 ± 0.59 | 92.29 ± 0.59 | OOM |

of the spectral augmentations compared to much simpler alternatives. Besides the major baselines mentioned above, other related ones are added to clearly and comprehensively benchmark our work. For MVGRL, we also compared its original PPR augmentation. For the node classification task, we use GCA (Zhu et al., 2021b), GMI (Peng et al., 2020), DGI (Velickovic et al., 2019), and SpCo (Liu et al., 2022a) as baselines. For the graph classification task, we use RGCL (Li et al., 2022) and GraphCL (You et al., 2020) as baselines. Detailed experimental configurations are in Appendix A.

**Evaluation Protocol.** We adopt the evaluation and split scheme from previous works (Veličković et al., 2019; Zhang et al., 2023; Lin et al., 2023). Each GNN encoder is trained on the entire graph with self-supervised learning. After training, we freeze the encoder and extract embeddings for all nodes or graphs. Finally, we train a simple linear classifier using the labels from the training/validation set and test it with the testing set. The accuracy of classification on the testing set shows how good the learned representations are. For the node classification task nodes are randomly divided into 10%/10%/80% for training, validation, and testing, and for graph classification datasets, graphs are randomly divided into 80%/10%/10% for training, validation, and testing.

## 6.2 EXPERIMENTAL RESULTS

We present the prediction accuracy of the node classification and graph classification tasks in Table 2 and Table 3, respectively. Our comparative analysis of graph augmentation for both node and graph classification reveals distinct performance trends. For node classification, DROPEDGE consistently achieves the best performance across multiple datasets and **CG-SSL** frameworks, demonstrating superior robustness and consistency. While ADDEDGE also achieves competitive accuracy, DROPEDGE stands out in this area. In graph classification, ADDEDGE frequently achieves the best performance across multiple datasets and **CG-SSL** frameworks, showing superior and more consistent results. In contrast, all the results from SPAN as well as GASSER and SpCo generally underperform relative to both DROPEDGE and ADDEDGE while also encountering scalability issues on larger datasets and suffering from a high overhead of training time.

## 6.3 ABLATION STUDY

To validate our findings, we conducted a series of ablation experiments on two exemplar datasets, CORA and MUTAG, representing node- and graph-level tasks, respectively. These ablation studies are crucial to rule out potential confounding variables, such as model architectures and hyperparameters, ensuring that our conclusions about the performance of **CG-SSL** are robust and comprehensive.

Table 3: Graph classification. Results of baselines with '†' are adopted directly from previous works. MVGRL+PPR is the original setting of MVGRL. The best results in each cell are highlighted by grey . The best results overall are highlighted with **bold and underline**. Metric is accuracy (%).

| Model | MUTAG | PROTEINS | NCI1 | IMDB-BINARY | IMDB-MULTI |
|---|---|---|---|---|---|
| GraphCL† | $86.80 \pm 1.34$ | $74.39 \pm 0.45$ | $77.87 \pm 0.41$ | $71.14 \pm 0.44$ | $48.58 \pm 0.67$ |
| RGCL† | $87.66 \pm 1.01$ | $75.03 \pm 0.43$ | $78.14 \pm 1.08$ | $71.85 \pm 0.84$ | $49.31 \pm 0.42$ |
| MVGRL + PPR | $90.00 \pm 5.40$ | $78.92 \pm 1.83$ | **$78.78 \pm 1.52$** | $71.40 \pm 4.17$ | **$52.13 \pm 1.42$** |
| MVGRL+ SPAN | $93.33 \pm 2.22$ | $79.81 \pm 2.45$ | $77.56 \pm 1.77$ | $75.00 \pm 1.09$ | $51.20 \pm 1.62$ |
| MVGRL+ DROPEDGE | $93.33 \pm 2.22$ | $78.92 \pm 1.33$ | $77.81 \pm 1.50$ | **$76.40 \pm 0.48$** | $51.46 \pm 3.02$ |
| MVGRL+ ADDEDGE | **$94.44 \pm 3.51$** | $81.25 \pm 3.43$ | $77.27 \pm 0.71$ | $74.00 \pm 2.82$ | $51.73 \pm 2.43$ |
| G-BT + SPAN | $90.00 \pm 6.47$ | **$80.89 \pm 3.22$** | $78.29 \pm 1.12$ | $65.60 \pm 1.35$ | $45.60 \pm 2.13$ |
| G-BT + DROPEDGE | $92.59 \pm 2.61$ | $77.97 \pm 0.42$ | $78.18 \pm 0.91$ | $73.33 \pm 1.24$ | $49.11 \pm 1.25$ |
| G-BT + ADDEDGE | $92.59 \pm 2.61$ | $80.64 \pm 1.68$ | $75.91 \pm 0.59$ | $73.33 \pm 1.24$ | $48.88 \pm 1.13$ |
| GRACE + SPAN | $90.00 \pm 4.15$ | $79.10 \pm 2.30$ | $78.49 \pm 0.79$ | $70.80 \pm 3.96$ | $47.73 \pm 1.71$ |
| GRACE + DROPEDGE | $88.88 \pm 3.51$ | $78.21 \pm 1.92$ | $76.93 \pm 1.14$ | $71.00 \pm 3.75$ | $47.46 \pm 3.02$ |
| GRACE + ADDEDGE | $92.22 \pm 4.44$ | $80.17 \pm 2.21$ | $79.97 \pm 2.35$ | $71.67 \pm 2.36$ | $49.86 \pm 4.09$ |
| BGRL + SPAN | $90.00 \pm 4.15$ | $79.28 \pm 2.73$ | $78.05 \pm 1.62$ | $72.40 \pm 2.57$ | $47.46 \pm 4.35$ |
| BGRL + DROPEDGE | $88.88 \pm 4.96$ | $76.60 \pm 2.21$ | $76.15 \pm 0.43$ | $71.60 \pm 3.31$ | $51.47 \pm 3.02$ |
| BGRL + ADDEDGE | $91.11 \pm 5.66$ | $79.46 \pm 2.18$ | $76.98 \pm 1.40$ | $72.80 \pm 2.48$ | $47.77 \pm 4.18$ |

**Number of Layers of GCN Encoder.** To assess the impact of model depth, we conducted both node-level and graph-level experiments using varying numbers of GCN encoder layers. This analysis is to rule out the possibility that model depth, rather than augmentation strategies, influences the claim. As expected, the results, detailed in Appendix E.1, show that deeper encoders generally lead to worse performance. This suggests that excessive model complexity may introduce noise or overfitting, diminishing the benefits of spectral information. Therefore, our conclusion still holds tightly.

**Type of GNN Encoder.** While we initially selected GCN to align with the common protocols in previous studies for a fair comparison, we also explored other GNN architectures to ensure our findings are not specific to GCN alone. To further validate our results, we conducted additional experiments using GAT (Veličković et al., 2019) for both node- and graph-level tasks, as well as GPS (Rampášek et al., 2024) for the graph-level task. As reported in Appendix E.2, the performance trends observed with GAT and GPS are consistent with those obtained using GCN. This consistency across different encoder types further supports our conclusion that simple edge perturbation strategies are sufficient, and that spectral augmentation does not significantly enhance performance, regardless of the type of GNN encoder applied.

## 7 THE INSIGNIFICANCE OF SPECTRAL CUES

Given the superior empirical performance of edge perturbations mentioned in Sec. 6, one may still argue whether it is a result of some spectral cues or not, as all the analyses mentioned are not direct evidence of the insignificance of the spectral information in the study. To clarify this, we have three questions to answer, **(1)** Can GNN encoders learn spectral information from augmented graphs produced edge perturbations? **(2)** Are spectrum in spectral augmentation necessary? **(3)** Is spectral information statistically a significant factor in the performance of edge perturbation? Given the questions, we conduct a series of experimental studies to answer them respectively in Sec. 7.1, 7.2 and Appendix E.4.

### 7.1 DEGENERATION OF THE SPECTRUM AFTER EDGE PERTURBATION (**EP**)

Here we want to conduct studies to answer the question of whether the GNN encoders applied can learn spectral information from the augmented graph views produced by **EP**. Therefore, we collect the spectrum of all augmented graphs ever produced along the way of the contrastive learning process of the best framework with the optimal parameter we have in this study, i.e., G-BT + **EP** with best drop rate $p$ or add rate $q$, and calculate the average one for each representative dataset in this study for both node- and graph-level tasks. We find that though the average spectrum of those original graphs is strikingly different, that of augmented graphs is quite similar for node- and graph-level tasks, respectively. This indicates a certain degree of degeneration of the spectra as they are no longer easy to separate after **EP**. Therefore, GNN encoders can hardly learn spectral information and properties between different original graphs from those augmented graph views. Note that, though we have

defined some context of frameworks, this result is generally only dependent on the augmentation methods. Due to the limited space, we will elaborate the node-level results in this section and postpone the graph-level ones in Appendix E.3, as they support the claim very consistently.

**Node-Level Analysis.** Here, we visualize the distributions of the average spectrum of graphs at the node level using histograms. The spectral distribution for each graph is represented by a sorted vector of its eigenvalues. When referring to the average spectrum, we mean the average over the eigenvalue vectors of each augmented graph. We plot the histograms of different spectra, normalized to show the probability density. Note that eigenvalues are constrained within the range [0, 2], as we adopted the commonly used symmetrical normalization. We analyze the spectral distributions of three node classification datasets: CORA, CITESEER, and COMPUTERS. We compare the average spectral properties of both original and augmented graphs. The augmentation method used is DROPEDGE, applied with optimal parameters identified for the G-BT method. The results of the visualization are presented in Fig. 2. By comparing the spectrum distributions of original graphs for the datasets in Fig. 2a, we can easily distinguish the spectra of the three datasets. This contrasts with the highly overlapped average spectra of all the datasets, indicating the degeneration mentioned. To support this claim, we also present the comparison of the spectra of original and augmented graphs on all three datasets in Fig. 2c, 2d, and 2e, respectively, to show the obvious changes after the edge perturbations.

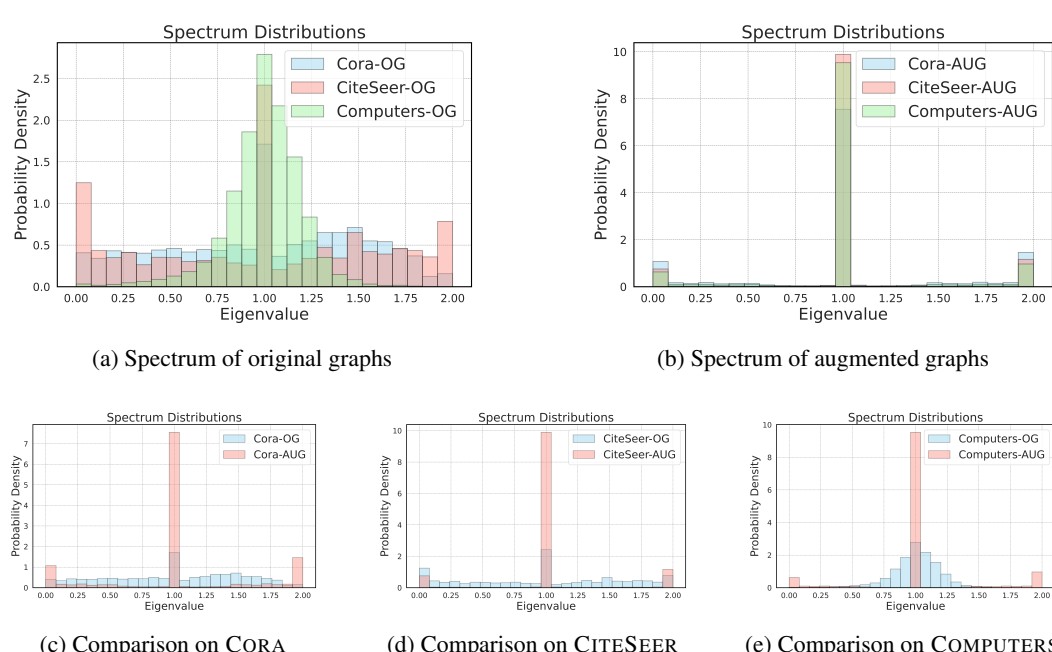

(a) Spectrum of original graphs       (b) Spectrum of augmented graphs

(c) Comparison on CORA    (d) Comparison on CITESEER    (e) Comparison on COMPUTERS

Figure 2: The spectrum distributions of graphs on different node classification datasets. CORA, CITESEER, and COMPUTERS are chosen as they are well representative of all the node classification datasets. OG means original graph and AUG means average augmented graphs. The augmentation method is DROPEDGE with the best parameter on G-BT method.

## 7.2 SPECTRAL PERTURBATION

To further destruct the spectral properties from model performance, we introduce *Spectral Perturbation Augmentor* (SPA) for finer-grained anatomy. SPA performs random edge perturbation with an empirically negligible ratio $r_{SPA}$ to transform the input graph $\mathcal{G}$ into a new graph $\mathcal{G}_{SPA}$, such that $\mathcal{G}$ and $\mathcal{G}_{SPA}$ are close to each other topologically, while being divergent in the spectral space. The spectral divergence $d_{SPA}$ between $\mathcal{G}$ and $\mathcal{G}_{SPA}$ is measured by the $L_2$-distance of the respective spectra. With properly chosen hyperparameters $r_{SPA}$ and $d_{SPA}$, we view the augmented graph $\mathcal{G}_{SPA}$ as a doppelganger of $\mathcal{G}$ that preserves most of the graph-proximity, with only spectral information eliminated.

**Spectral perturbation on spectral augmentation baselines.** SPAN, being a state-of-the-art spectral augmentation algorithm, demonstrated the correlation between graph spectra and model performance

through designated perturbation on spectral priors. However, the effectiveness of simple edge perturbation motivated us to further investigate whether such a relationship is causational.

Specifically, for each pair of SPAN augmented graphs $\mathcal{G}^1, \mathcal{G}^2$, we further augment them into $\mathcal{G}^1_{SPA}, \mathcal{G}^2_{SPA}$ with our proposed SPA augmentor. The SPA-augmented training is performed under the same setup as SPAN, with graphs being SPA-augmented graphs $\mathcal{G}_{SPA}$. Experiment results in Fig 3 show that the effectiveness of graph augmentation can be preserved and, in some cases, improved, even if the spectral information is destroyed.

SPAN, along with other spectral augmentation algorithms, can be formulated as an optimization on a parameterized 2-step generative process:

$$s_{SPAN} \sim p_\theta \left( \boldsymbol{S}_{SPAN} \mid \boldsymbol{\mathcal{G}}_0 \right), \qquad \mathcal{G}_{SPAN} \sim p_\phi \left( \boldsymbol{\mathcal{G}}_{SPAN} \mid \boldsymbol{S}_{SPAN} \right) \tag{5}$$

Given the property that $\mathcal{G}_{SPA}$ is topologically close to $\mathcal{G}_{SPAN}$ and the performance function $\mathrm{P} = f \left( \mathcal{G} \right), \lim_{\mathcal{G} \to \mathcal{G}_{SPAN}} \mathrm{P} \left( \mathcal{G} \right) = \mathrm{P} \left( \mathcal{G}_{SPAN} \right)$, which indicates the continuity around $\mathcal{G}_{SPAN}$, we make a reasonable assertion that $\mathcal{G}_{SPA}$ comes from the same distribution as $\mathcal{G}_{SPAN}$. However, with their spectral space being enforced to be distant, $\mathcal{G}_{SPA}$ is almost impossible to be sampled from the same spectral augmentation generative process:

$$d_{SPA} \to \infty \implies p_\theta \left( s_{SPA} \mid \boldsymbol{\mathcal{G}}_0 \right) \to 0 \implies p_{\theta,\phi} \left( \mathcal{G}_{SPA} \mid \boldsymbol{\mathcal{G}}_0 \right) \to 0 \tag{6}$$

Although the constrained generative process in Eq. 5 does indicate some extent of causality between spectral distribution $\boldsymbol{S}$ and the spectral-augmented graph distribution $\boldsymbol{\mathcal{G}}_{SPAN}$, our experiment challenges a more essential and fundamental aspect of such reasoning: such causality exists upon pre-defined generative processes, which does not intrinsically exist in the graph distributions. Even worse, such constrained generative process is incapable of modeling the full distribution of $\boldsymbol{\mathcal{G}}_{SPAN}$ itself. In our experiment setup, all $\mathcal{G}_{SPA}$ serve as strong counter examples.

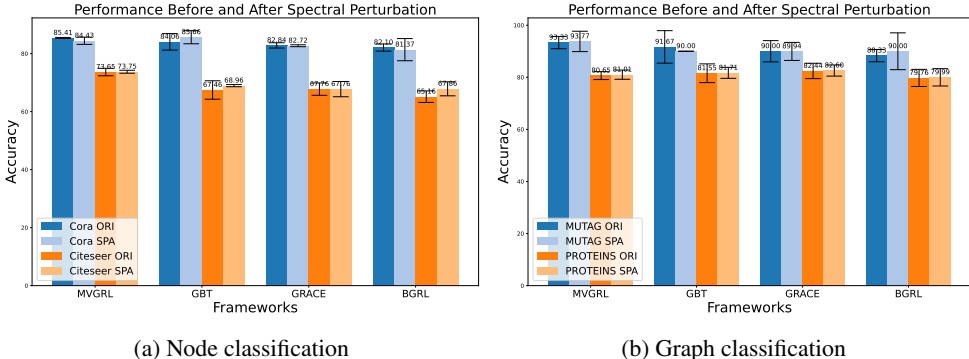

(a) Node classification          (b) Graph classification

Figure 3: Comparison of SPAN performance before and after applying SPA. After severely disrupting the spectral, the performance of SPAN is still comparable to that of the original version.

## 8 CONCLUSION

In this study, we investigate the effectiveness of spectral augmentation in contrast-based graph self-supervised learning (**CG-SSL**) frameworks to answer the question: *Are spectral augmentations necessary in CG-SSL*? Our findings indicate that spectral augmentation does not significantly enhance learning efficacy. Instead, simpler edge perturbation techniques, such as random edge dropping for node-level tasks and random edge adding for graph-level tasks, not only compete well but often outperform spectral augmentations. To be specific, we demonstrate that the benefits of spectral augmentation diminish with shallower networks, and edge perturbations yield superior performance in both node- and graph-level classification tasks. Additionally, GNN encoders struggle to learn spectral information from augmented graphs, and perturbing edges to alter spectral characteristics does not degrade model performance. These results challenge the current emphasis on spectral augmentation, advocating for more straightforward and effective edge perturbation techniques in **CG-SSL**, potentially reshaping the understanding and implementation of graph self-supervised learning.

**Ethics Statement**  To the authors' best knowledge, no major ethics issues in this submission.

**Reproducibility Statement**  We have made efforts to ensure the reproducibility of our work:

- **Datasets:** All datasets used in this study are publicly available through the PyTorch Geometric (PyG) library[1]. The statistics of node-level and graph-level datasets are detailed in Tables 4 and 5 respectively.
- **Implementation:** Our CG-SSL framework implementation is based on the work of Zhu et al. (2021a)[2]. We will open-source our code in the near future to facilitate reproducibility.

No additional data processing steps were required beyond those inherent in the PyG library. Detailed model architectures and evaluation protocols are provided in the Sec. 6.1 and Appendix A.

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

Table 4: Statistics of node classification datasets

| Dataset | #Nodes | #Edges | #Features | #Classes |
|---|---|---|---|---|
| CORA | 2,708 | 5,429 | 1,433 | 7 |
| CITESEER | 3,327 | 4,732 | 3,703 | 6 |
| PUBMED | 19,717 | 44,338 | 500 | 3 |
| COMPUTERS | 13,752 | 245,861 | 767 | 10 |
| PHOTO | 7,650 | 119,081 | 745 | 8 |
| COAUTHOR-CS | 18,333 | 81,894 | 6,805 | 15 |
| COAUTHOR-PHY | 34,493 | 247,962 | 8,415 | 5 |

Table 5: Statistics of node classification datasets

| Dataset | #Avg. Nodes | #Avg. Edges | # Graphs | #Classes |
|---|---|---|---|---|
| MUTAG | 17.93 | 19.71 | 188 | 2 |
| PROTEINS | 39.06 | 72.82 | 1,113 | 2 |
| NCI1 | 29.87 | 32.30 | 4110 | 2 |
| IMDB-BINARY | 19.8 | 96.53 | 1,000 | 2 |
| IMDB-MULTI | 13.0 | 65.94 | 1,500 | 5 |

## A  DATASET AND TRAINING CONFIGURATION

**Datasets.** The node classification datasets used in this paper include the CORA, CITESEER, and PUBMED citation networks (Kipf & Welling, 2016), as well as the PHOTO and COMPUTERS co-purchase networks (Shchur et al., 2018). Additionally, we use the COAUTHOR-CS and COAUTHOR-PHY co-author relationship networks. The statistics of node-level datasets are present in Table 4. The graph classification datasets include: The MUTAG dataset, which features seven types of graphs derived from 188 mutagenic compounds; the NCI1 dataset, which contains compounds tested for their ability to inhibit human tumor cell growth; the PROTEINS dataset, where nodes correspond to secondary structure elements connected if they are adjacent in 3D space; and the IMDB-BINARY and IMDB-MULTI movie collaboration datasets, where graphs depict interactions among actors and actresses, with edges denoting their collaborations in films. These movie graphs are labeled according to their genres. The statistics of graph-level datasets are present in Table 5. All datasets can be accessed through PyG library [3]. All experiments are conducted using 8 NVIDIA A100 GPU.

**Training configuration.** For each **CG-SSL** framework, we implement it based on (Zhu et al., 2021a) [4]. We use the following hyperparameters: the learning rate is set to $5 \times 10^{-4}$, and the node hidden size is set to $512$, the number of GCN encoder layer is set $\in \{1, 2\}$. For all node classification datasets, training epochs are set $\in \{50, 100, 150, 200, 400, 1000\}$, and for all graph classification datasets, training epochs are set $\in \{20, 40, ..., 200\}$. To achieve performance closer to the global optimum, we use randomized search to determine the optimal probability of edge perturbation and SPAN perturbation ratio. For CORA and CITESEER the search is conducted one hundred times, and for all other datasets, it is conducted twenty times. For all graph classification datasets, the batch size is set to 128.

## B  PRELIMINARIES OF GRAPH SPECTRUM AND SPAN

Given a graph $\mathcal{G} = (\mathbf{A}, \mathbf{X})$ with adjacency matrix $\mathbf{A}$ and feature matrix $\mathbf{X}$, we introduce some fundamental concepts related to the graph spectrum.

**Laplacian Matrix Spectrum** The Laplacian matrix $\mathbf{L}$ of a graph is defined as:

$$\mathbf{L} = \mathbf{D} - \mathbf{A}$$

where $\mathbf{D}$ is the degree matrix, a diagonal matrix where each diagonal element $D_{ii}$ represents the degree of vertex $i$. The eigenvalues of the Laplacian matrix, known as the Laplacian spectrum, are

---

[3] https://pytorch-geometric.readthedocs.io/en/latest/modules/datasets.html

[4] https://github.com/PyGCL/PyGCL

crucial in understanding the graph's structural properties, such as its connectivity and the number of spanning trees (Chung, 1997).

**Normalized Laplacian Spectrum** The normalized Laplacian matrix $\mathbf{L}_{\text{norm}}$ is given by:

$$\mathbf{L}_{\text{norm}} = \mathbf{D}^{-1/2}\mathbf{L}\mathbf{D}^{-1/2}$$

The eigenvalues of the normalized Laplacian matrix, referred to as the normalized Laplacian spectrum, are often used in spectral clustering (Von Luxburg, 2007) and other applications where normalization is necessary to account for varying vertex degrees.

**SPAN** The core assumption of SPAN is to maximize the consistency of the representations of two views with a large spectrum distance, thereby filtering out edges sensitive to the spectrum, such as edges between clusters. By focusing on more stable structures relative to the spectrum, the objective of SPAN can be formulated as:

$$\max_{\mathcal{T}_1, \mathcal{T}_2 \in \mathcal{S}} \left\| \text{eig}\left(\mathbf{L}_1\right) - \text{eig}\left(\mathbf{L}_2\right) \right\|_2^2 \tag{7}$$

where the transformations $\mathcal{T}_1$ and $\mathcal{T}_2$ convert $\mathbf{A}$ to $\mathbf{A}_1$ and $\mathbf{A}_2$, respectively, producing the normalized Laplacian matrices $\mathbf{L}_1$ and $\mathbf{L}_2$. Here, $\mathcal{S}$ represents the set of all possible transformations, and the graph spectrum can be calculated by $\text{eig}(\mathbf{L})$.

## C  OBJECT FUNCTION OF GCL FRAMEWORK

Here we briefly introduce the object functions of the four **CG-SSL** frameworks used in this paper, for a more detailed discussion about object functions including other graph contrastive learning and graph self-supervised learning frameworks which can refer to the survey papers (Xie et al., 2022; Wu et al., 2021; Liu et al., 2022b). We use the following notations:

- $p_\phi$: Projection head parameterized by $\phi$.
- $\mathbf{h}_i, \mathbf{h}_j$: Representations of the graph nodes.
- $\mathbf{h}'_n$: Representations of negative sample nodes.
- $\mathcal{P}$: Distribution of positive sample pairs.
- $\widetilde{\mathcal{P}}^N$: Distribution of negative sample pairs.
- $\mathcal{B}$: Set of nodes in a batch.
- $\mathbf{H}^{(1)}, \mathbf{H}^{(2)}$: Node representation matrices of two views.

GRACE uses the InfoNCE loss to maximize the similarity between positive pairs and minimize the similarity between negative pairs. InfoNCE loss encourages representations of positive pairs (generated from the same node via data augmentation) to be similar while pushing apart the representations of negative pairs (from different nodes). The loss function $\mathcal{L}_{\text{NCE}}$ denotes as:

$$\mathcal{L}_{\text{NCE}}\left(p_\phi\left(\mathbf{h}_i, \mathbf{h}_j\right)\right) = -\mathbb{E}_{\mathcal{P} \times \widetilde{\mathcal{P}}^N}\left[\log \frac{e^{p_\phi(\mathbf{h}_i, \mathbf{h}_j)}}{e^{p_\phi(\mathbf{h}_i, \mathbf{h}_j)} + \sum_{n \in N} e^{p_\phi(\mathbf{h}_i, \mathbf{h}'_n)}}\right] \tag{8}$$

MVGRL employs the Jensen-Shannon Estimator (JSE) for contrastive learning, which focuses on the mutual information between positive pairs and negative pairs.JSE maximizes the mutual information between positive pairs and minimizes it for negative pairs, thus improving the representations' alignment and uniformity. The loss function $\mathcal{L}_{\text{JSE}}$ denotes as:

$$\mathcal{L}_{\text{JSE}}\left(p_\phi\left(\mathbf{h}_i, \mathbf{h}_j\right)\right) = \mathbb{E}_{\mathcal{P} \times \widetilde{\mathcal{P}}}\left[\log\left(1 - p_\phi\left(\mathbf{h}_i, \mathbf{h}'_j\right)\right)\right] - \mathbb{E}_{\mathcal{P}}\left[\log\left(p_\phi\left(\mathbf{h}_i, \mathbf{h}_j\right)\right)\right] \tag{9}$$

BGRL utilizes a loss similar to BYOL, which does not require negative samples. It uses two networks, an online network and a target network, to predict one view from the other:

$$\mathcal{L}_{\text{BYOL}}\left(p_\phi\left(\mathbf{h}_i, \mathbf{h}_j\right)\right) = \mathbb{E}_{\mathcal{P} \times \mathcal{P}}\left[2 - 2 \cdot \frac{\left[p_\phi\left(\mathbf{h}_i\right)\right]^T \mathbf{h}_j}{\|p_\phi\left(\mathbf{h}_i\right)\| \, \|\mathbf{h}_j\|}\right] \tag{10}$$

G-BT applies the Barlow Twins' loss to reduce redundancy in the learned representations, thereby ensuring better generalization:

$$
\begin{aligned}
\mathcal{L}_{\text{BT}}\left(\mathbf{H^{(1)}}, \mathbf{H^{(2)}}\right) = {} & \mathbb{E}_{\mathcal{B} \sim \mathcal{P}|\mathcal{B}|}\left[\sum_a\left(1 - \frac{\sum_{i \in \mathcal{B}} \mathbf{H}_{ia}^{(1)} \mathbf{H}_{ia}^{(2)}}{\left\|\mathbf{H}_{ia}^{(1)}\right\| \left\|\mathbf{H}_{ia}^{(2)}\right\|}\right)^2\right. \\
& \left. + \lambda \sum_a \sum_{b \neq a}\left(\frac{\sum_{i \in \mathcal{B}} \mathbf{H}_{ia}^{(1)} \mathbf{H}_{ib}^{(2)}}{\left\|\mathbf{H}_{ia}^{(1)}\right\| \left\|\mathbf{H}_{ib}^{(2)}\right\|}\right)^2\right].
\end{aligned}
\tag{11}
$$

## D  THEORETICAL GAPS IN SPECTRAL AUGMENTATION FOR GRAPH LEARNING

A significant gap exists between the theoretical foundations of spectral methods (Liu et al., 2022a; Yang et al., 2023) and their practical application in graph learning. Applying spectral theory to graph learning is often non-trivial, as it typically requires several simplifying assumptions that may not hold in real-world scenarios. This disconnect is evident in the underlying motivations of many spectral-based self-supervised learning (**CG-SSL**) methods. While spectral techniques aim to harness the eigenvalues and eigenvectors of graph Laplacians, their direct application to SSL tasks often leads to assumptions that are challenging to justify in practice (Liu et al., 2022a; Yang et al., 2023).

For instance, Theorem 1 in SpCo (Liu et al., 2022a) posits an upper bound on the InfoNCE loss in terms of the $L_2$ distance between the eigenvalues of the original and augmented graphs, moderated by adaptive weights:

$$\mathcal{L}_{\text{InfoNCE}} \leq \frac{1+N}{2} \sum_i \theta_i \left[2 - (\lambda_i - \gamma_i)^2\right]. \tag{12}$$

However, this relationship is relatively loose, resting on assumptions such as using a GCN encoder without activation layers. Furthermore, while InfoNCE is a widely used contrastive learning objective, its upper bound guarantees performance only within the specific contrastive training setup. This does not necessarily reflect the quality of the learned representations themselves, nor does it imply that these representations will perform well on downstream tasks like node classification. Thus, while the theorem provides useful theoretical insights, its direct relevance to practical graph learning tasks remains limited.

## E  MORE EXPERIMENTS

### E.1  EFFECT OF NUMBERS OF GCN LAYERS

We explore the impact of GCN depth on accuracy by testing GCNs with 4, 6, and 8 layers, using our edge perturbation methods alongside SPAN baselines. Experiments were conducted with the GRACE and G-BT frameworks on the Cora dataset for node classification and the MUTAG dataset for graph classification. Each configuration was run three times, with the mean accuracy and standard deviation reported.

Overall, deeper GCNs (6 and 8 layers) tend to perform worse across both tasks, reinforcing the observation that deeper architectures, despite their theoretical expressive power, may negatively impact the quality of learned representations. The results are summarized in Tables 6 and 7.

Table 6: Impact of GCN depth on node classification task on the CORA dataset. The best result of each column is in grey . Metric is accuracy (%).

| MODEL | 4 | 6 | 8 |
|---|---|---|---|
| GBT+DROPEDGE | 83.53± 1.48 | 82.06± 3.45 | 80.88± 1.38 |
| GBT +ADDEDGE | 81.99± 0.79 | 79.04± 1.59 | 79.41± 1.98 |
| GBT+SPAN | 80.39± 2.17 | 81.25± 1.67 | 79.41± 1.87 |
| GRACE+DROPEDGE | 82.35± 1.08 | 82.47± 1.35 | 81.74± 2.42 |
| GRACE +ADDEDGE | 79.17 ±1.35 | 78.80± 0.96 | 81.00± 0.17 |
| GRACE+SPAN | 80.15± 0.30 | 80.15± 0.79 | 75.98± 1.54 |

Table 7: Impact of GCN depth on graph classification task on the MUTAG dataset. The best result of each column is in grey . Metric is accuracy (%).

| MODEL | 4 | 6 | 8 |
|---|---|---|---|
| GBT+DROPEDGE | 90.74 ± 2.61 | 88.88 ± 4.53 | 88.88 ± 7.85 |
| GBT +ADDEDGE | 94.44 ± 0.00 | 94.44 ± 4.53 | 94.44 ± 4.53 |
| GBT+SPAN | 94.44 ± 4.53 | 92.59 ± 2.61 | 90.74 ± 2.61 |
| GRACE+DROPEDGE | 94.44 ± 0.00 | 90.74 ± 2.61 | 90.74 ± 2.61 |
| GRACE +ADDEDGE | 92.59 ± 5.23 | 94.44 ± 4.53 | 94.44 ± 0.00 |
| GRACE+SPAN | 90.74 ± 2.61 | 90.74 ± 5.23 | 88.88 ± 7.85 |

## E.2 EFFECT OF GNN ENCODER

To further validate the generality of our approach, we conducted additional experiments using different GNN encoders. For the node classification task, we evaluated the CORA dataset with GAT as the encoder, while for the graph classification task, we performed experiments on the MUTAG dataset using both GAT and GPS as encoders.

The results, presented in Tables 8 and 9, are shown alongside the results obtained with GCN encoders. These findings demonstrate that our simple edge perturbation method consistently outperforms the baselines, regardless of the choice of the encoder. This confirms that our conclusions hold across different encoder architectures, underscoring the robustness and effectiveness of the proposed approach.

## E.3 GRAPH-LEVEL ANALYSIS FOR DEGENERATION OF THE SPECTRUM AFTER **EP** (SEC. 7.1 CONT.)

For graph-level analysis, we basically follow the settings mentioned above in node-level one. The only difference from the node-level task is that we have multiple original graphs with various numbers of nodes, leading to the inconsistent dimensions of the vector of the eigenvalues. Therefore, to provide a more detailed comparison of spectral properties at the graph level, we employ Kernel Density Estimation (KDE) (Parzen, 1962) to interpolate and smooth the distributions of eigenvalues. We compare two groups of graph spectra. Each group's spectra are processed to compute their KDEs, and the mean and standard deviation of these KDEs are calculated.

We analyze the spectral distributions of two node classification datasets: MUTAG and PROTEINS. We compare the average spectral properties of both original and augmented graphs. The augmentation method used is ADDEDGE as it is the better among two **EP** methods, applied with optimal add rate identified for the G-BT method.

Like the results in node-level analysis, in Fig. 4a and 4b, we witness the obvious difference between the average spectra of original graphs while the significant overlap between those of augmented graphs, especially if pay attention to the overlapping of the area created by the standard deviation of KDEs. Again, this contrast is not trivial because of the striking mismatch between the average spectra of original and augmented graphs in both datasets, as presented in Fig. 4c and 4d.

Table 8: Accuracy of node classification with different GNN encoders on CORA dataset. The best result of each column is in grey . Metric is accuracy (%).

| MODEL | GCN | GAT |
|---|---|---|
| MVGRL+SPAN | $84.57 \pm 0.22$ | $82.90 \pm 0.86$ |
| MVGRL+DROPEDGE | $84.31 \pm 1.95$ | $83.21 \pm 1.41$ |
| MVGRL +ADDEDGE | $83.21 \pm 1.65$ | $83.33 \pm 0.17$ |
| GBT+SPAN | $82.84 \pm 0.90$ | $83.47 \pm 0.39$ |
| GBT + DROPEDGE | $84.19 \pm 2.07$ | $84.06 \pm 1.05$ |
| GBT + ADDEDGE | $85.78 \pm 0.62$ | $81.49 \pm 0.45$ |
| GRACE + SPAN | $82.84 \pm 0.91$ | $82.74 \pm 0.47$ |
| GRACE + DROPEDGE | $84.19 \pm 2.07$ | $82.84 \pm 2.58$ |
| GRACE + ADDEDGE | $85.78 \pm 0.62$ | $82.84 \pm 1.21$ |
| BGRL + SPAN | $83.33 \pm 0.45$ | $82.59 \pm 0.79$ |
| BGRL + DROPEDGE | $83.21 \pm 3.29$ | $80.88 \pm 1.08$ |
| BGRL + ADDEDGE | $81.49 \pm 1.21$ | $82.23 \pm 2.00$ |

Table 9: Accuracy of graph classification with different GNN encoders on MUTAG dataset. The best result of each column is in grey . Metric is accuracy (%).

| MODEL | GCN | GAT | GPS |
|---|---|---|---|
| MVGRL+SPAN | $93.33 \pm 2.22$ | $96.29 \pm 2.61$ | $94.44 \pm 0.00$ |
| MVGRL+DROPEDGE | $93.33 \pm 2.22$ | $92.22 \pm 3.68$ | $96.26 \pm 5.23$ |
| MVGRL +ADDEDGE | $94.44 \pm 3.51$ | $94.44 \pm 6.57$ | $95.00 \pm 5.24$ |
| GBT+SPAN | $90.00 \pm 6.47$ | $94.44 \pm 4.53$ | $90.74 \pm 5.23$ |
| GBT + DROPEDGE | $92.59 \pm 2.61$ | $94.44 \pm 4.53$ | $94.44 \pm 4.53$ |
| GBT + ADDEDGE | $92.59 \pm 2.61$ | $92.59 \pm 2.61$ | $94.44 \pm 4.53$ |
| GRACE + SPAN | $90.00 \pm 4.15$ | $96.29 \pm 2.61$ | $92.59 \pm 2.61$ |
| GRACE + DROPEDGE | $88.88 \pm 3.51$ | $94.44 \pm 0.00$ | $94.44 \pm 4.53$ |
| GRACE + ADDEDGE | $92.22 \pm 4.22$ | $96.29 \pm 2.61$ | $94.44 \pm 0.00$ |
| BGRL + SPAN | $90.00 \pm 4.15$ | $94.44 \pm 4.53$ | $94.44 \pm 0.00$ |
| BGRL + DROPEDGE | $88.88 \pm 4.96$ | $90.74 \pm 4.54$ | $92.59 \pm 5.23$ |
| BGRL + ADDEDGE | $91.11 \pm 5.66$ | $96.29 \pm 2.61$ | $96.29 \pm 2.61$ |

## E.4 RELATIONSHIP BETWEEN SPECTRAL CUES AND PERFORMANCE OF EP

Based on the findings obtained from Sec 7.1, it is very likely that spectral information can not be distinguishable enough for good representation learning on the graph. But to more directly answer the question of whether spectral cues and information play an important role in the learning performance of **EP**, we continue to conduct a statistical analysis to evaluate the influence of various factors on the learning performance. The results turn out to be consistent with our claim that spectral cues are insignificant aspects of outstanding performance on accuracy observed in Sec. 6.

### E.4.1 STATISTICAL ANALYSES ON KEY FACTORS ON PERFORMANCE OF EP

From a statistical angle, we have a few dimensions of factors that can possibly influence learning performance, like the parameters of **EP** (i.e. drop rate $p$ in DROPEDGE or add rate $q$ in ADDEDGE) as well as potential spectral cues lying in the argument graphs. Therefore, to rule out the possibility that spectral cues and information are significant, comparisons are conducted on the impact of the parameters of **EP** in the augmentations versus:

1. The average $L_2$-distance between the spectrum of the original graph (OG) and that of each augmented graph (AUG) which is introduced by **EP** augmentations, denoted as OG-AUG.

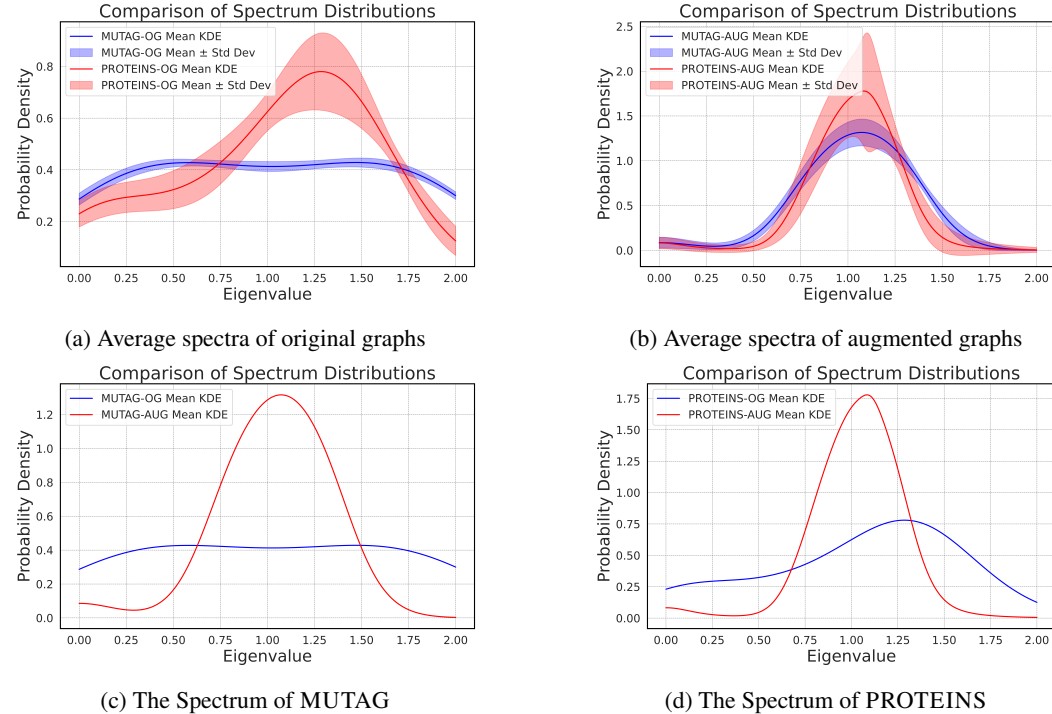

(a) Average spectra of original graphs

(b) Average spectra of augmented graphs

(c) The Spectrum of MUTAG

(d) The Spectrum of PROTEINS

Figure 4: The spectrum distributions of graphs on different graph classification datasets. MUTAG and PROTEINS are chosen as they are well representative of all the node classification datasets. OG means original graph and AUG means augmented graph. The augmentation method is ADDEDGE with the best parameter on G-BT method.

2. The average $L_2$-distance between the spectra of a pair of augmented graphs appearing in the same learning epoch when having a two-way contrastive learning framework, like G-BT, denoted as AUG-AUG.

Two statistical analyses have been carried out to argue that the former is a more critical determinant and a more direct cause of the model efficacy. Each analysis was chosen for its ability to effectively dissect and compare the impact of edge perturbation parameters versus spectral changes.

Due to the high cost of calculating the spectrum of all AUGs in each epoch and the stability of the spectrum of the node-level dataset (as the original graph is fixed in the experiment), we perform this experiment on the contrastive framework and augmentation methods with the best performance in the study, i.e. G-BT with DROPEDGE on node-level classification. Also, we choose the small datasets, CORA for analysis. Note that the smaller the graph, the higher the probability that the spectrum distance has a significant influence on the graph topology.

**Analysis 1: Polynomial Regression.** Polynomial regression was utilized to directly model the relationship between the test accuracy of the model and the average spectral distances introduced by **EP**. This method captures the linear, or non-linear influences that these spectral distances may exert on the learning outcomes, thereby providing insight into how different parameters affect model performance.

Table 10: Polynomial regression of node-level accuracy over drop rate $p$ in DROPEDGE, average spectral distance between OG and AUG (OG-AUG), and average spectral distance between AUG pairs (AUG-AUG). The method is G-BT and the dataset is CORA. The best results are in grey.

| Order of the regression | Regressor | R-squared ↑ | Adj. R-squared ↑ | F-statistic ↑ | P-value ↓ |
|---|---|---|---|---|---|
| 1 (i.e. linear) | Drop rate $p$ | 0.628 | 0.621 | 81.12 | 6.94e-12 |
| | OG-AUG | 0.388 | 0.375 | 30.45 | 1.35e-06 |
| | AUG-AUG | 0.338 | 0.325 | 24.55 | 9.39e-06 |
| 2 (i.e. quadratic) | Drop rate $p$ | 0.844 | 0.837 | 126.9 | 1.14e-19 |
| | OG-AUG | 0.721 | 0.709 | 60.78 | 9.23e-14 |
| | AUG-AUG | 0.597 | 0.580 | 34.88 | 5.16e-10 |

The polynomial regression analysis in Table 10 highlights that the drop rate $p$ is the primary factor influencing model performance, showing strong and significant linear and non-linear relationships with test accuracy. In contrast, both the OG-AUG and AUG-AUG spectral distances have relatively minor impacts on performance, indicating that they are not significant determinants of the model's efficacy.

**Analysis 2: Instrumental Variable Regression.** To study the causal relationship, we perform an Instrumental Variable Regression (IVR) to rigorously evaluate the influence of spectral information and edge perturbation parameters on the performance of **CG-SSL** models. Specifically, we employ a Two-Stage Least Squares (IV2SLS) method to address potential endogeneity issues and obtain unbiased estimates of the causal effects.

In IV2SLS analysis, we define the variables as follows:

- **Y (Dependent Variable):** The outcome we aim to explain or predict, which in this case is the performance of the SSL model.

- **X (Explanatory Variable):** The variable that we believe directly influences Y. It is the primary factor whose effect on Y we want to measure.

- **Z (Instrumental Variable):** A variable that is correlated with X but not with the error term in the Y equation. It helps to isolate the variation in X that is exogenous, providing a means to obtain unbiased estimates of X's effect on Y.

In this specific experiment, we conduct four separate regressions to compare the causal effects of these factors:

1. **(X = AUG-AUG, Z = Parameter):** Examines the relationship where the spectral distance between augmented graphs (AUG-AUG) is the explanatory variable (X) and edge perturbation parameters are the instrument (Z).

2. **(X = Parameter, Z = AUG-AUG):** Examines the relationship where the edge perturbation parameters are the explanatory variable (X) and the spectral distance between augmented graphs (AUG-AUG) is the instrument (Z).

3. **(X = OG-AUG, Z = Parameter):** Examines the relationship where the spectral distance between the original and augmented graphs (OG-AUG) is the explanatory variable (X) and edge perturbation parameters are the instrument (Z).

4. **(X = Parameter, Z = OG-AUG):** Examines the relationship where the edge perturbation parameters are the explanatory variable (X) and the spectral distance between the original and augmented graphs (OG-AUG) is the instrument (Z).

The IV2SLS regression results for the node-level task in Table 11 indicate that the edge perturbation parameters are more significant determinants of model performance than spectral distances. Specifically, when the spectral distance between augmented graphs (AUG-AUG) is the explanatory variable (X) and drop rate $p$ are the instrument (Z), the model explains 34.1% of the variance in performance (R-squared = 0.341). Conversely, when the roles are reversed (X = $p$, Z = AUG-AUG), the model explains 61.1% of the variance (R-squared = 0.611), indicating a stronger influence of edge perturbation parameter $p$. A similar conclusion can be made when comparing OG-AUG and $p$.

Table 11: IV2SLS regression results for the node-level task. The parameter $p$ refers to the drop rate in DROPEDGE. The experiment comes in pairs for each pair of variables and the better result is marked in grey .

| Variable settings | R-squared ↑ | F-statistic ↑ | Prob (F-statistic) ↓ |
|---|---|---|---|
| ($\mathbf{X}$ = AUG-AUG, $\mathbf{Z} = p$) | 0.341 | 45.77 | 1.68e-08 |
| ($\mathbf{Z} = p$ ,$\mathbf{Z}$ = AUG-AUG) | 0.611 | 47.85 | 9.85e-09 |
| ($\mathbf{X}$ = OG-AUG, $\mathbf{Z} = p$) | 0.250 | 40.22 | 7.51e-08 |
| ($\mathbf{X} = p$, $\mathbf{Z}$ = OG-AUG) | 0.606 | 41.27 | 5.62e-08 |

**Summary of Regression Analyses**    The analyses distinctly show that the direct edge perturbation parameters have a consistently stronger and more significant impact on model performance than the two types of spectral distances that serve as a reflection of spectral information. The results support the argument that while spectral information might have contributed to model performance, its significance is extremely limited and the parameters of the **EP** methods themselves are more critical determinants.

