# OpenReview forum: "Are spectral augmentations necessary in contrast-based graph self-supervised learning?"
_ICLR.cc/2025/Conference — ICLR 2025 Conference Withdrawn Submission_

### Official Review · Reviewer_kJTT · 2024-11-02

**Soundness:** 2
**Presentation:** 3
**Contribution:** 2
**Rating:** 3
**Confidence:** 4

**Summary:**

This paper aims to discuss spectral augmentations for graph contrastive learning, with a primary focus on edge augmentation. It is important to note that adding or deleting edges affects the graph structure, which consequently impacts the eigenvalues and eigenvectors of the Laplacian matrix. The DropEdge method in graph contrastive learning has been well studied previously.

**Strengths:**

- It provides extensive empirical evidence through experiments on various datasets and frameworks, demonstrating the effectiveness of the proposed edge augmentation methods.
- The paper shows that the edge augmentation methods are effective across different CG-SSL frameworks, indicating broad applicability and versatility.

**Weaknesses:**

- The relationship between spectral augmentations and edge augmentation is not thoroughly analyzed theoretically in the paper.
- The improvement in performance primarily relies on a very simple edge augmentation, which is not a novel concept in graph contrastive learning.
- The main motivation of the paper is based on empirical studies rather than being well-discussed from a theoretical perspective.
- The methods of CCA-SSA and M-ILBO are not adequately discussed.

**Questions:**

- GRACE and other works by Zhu et al. provide significant insights into edge augmentation. How does your work differ from theirs?
- Some of the comparison baselines are not sufficiently novel. Why did you not compare your method to CCA-SSA and M-ILBO?
- Does the main paper rely heavily on empirical studies? Could you elaborate on the theoretical analysis?

---

### Official Review · Reviewer_XEMh · 2024-11-03

**Soundness:** 3
**Presentation:** 2
**Contribution:** 2
**Rating:** 5
**Confidence:** 4

**Summary:**

This paper explores the necessity of spectral augmentations in contrast-based graph self-supervised learning (CG-SSL). While spectral augmentation, i.e., modifying graph properties like eigenvalues, has been proposed as an alternative to naive edge perturbation, the authors empirically analyze its effectiveness in practice. In particular, they find that random edge perturbations such as ADDEDGE or DROPEDGE consistently outperform spectral methods, while also offering significantly lower computational costs. Finally, based on the observation that random edge perturbation leads to trivial degeneration of graph spectra, the authors hypothesize that graph contrastive learning does not necessarily rely on spectral information for producing meaningful embeddings.

**Strengths:**

This paper studies an important problem and presents a systematic empirical study. The noteworthy contributions of this work include
+ Rigorous comparison of random edge perturbations against SPAN (a popular spectral aug. method)
+ Experiments with a suite of GCL methods, GNN architectures and node/graph classification benchmarks
+ For the large scope of this empirical study, the paper is well-organized

**Weaknesses:**

While the paper will be of broad interest to the graph learning community, there are a number of aspects that need more insights to make it more effective.
+ In Fig.1, the superior performance of shallow GNNs is interpreted as an evidence for the claim that incorporating spectral information can affect the quality of the learned representations. However, there are a number of known issues in training deeper graph neural networks such as over-smoothing, difficulty in preserving locality, noise propagation etc. (see references like Bag of Tricks for Training Deeper Graph Neural Networks: A Comprehensive Benchmark Study, IEEE TPAMI 2022). This has led to the use of mitigation strategies like skip connections, layer normalization or even spectral GNNs. While the utility of the inductive biases even in shallow GNNs is clear from results (and also other existing studies), extending that observation to the claim on spectral information hurting representation quality needs additional empirical and theoretical justification.
+ While the performance improvements of edge perturbation over SPAN is apparent from the results in Table 2 and Table 3, it is not clear why DROPEDGE is significantly superior to ADDEDGE in node classification, while they are comparable (ADDEDGE is sometimes even superior) in graph classification. Can the authors explain this behavior?
+ The analysis in Figure 2 shows that DROPEDGE augmentation leads to a degeneration in the resulting graph spectra. Why does GCL with such samples lead to superior representations? While it is clear that the spectral content is not informative in this scenario, what would explain the behavior of contrastive learning? How would this argument relate to existing theoretical frameworks for self-supervised learning (Provable guarantees for self- supervised deep learning with spectral contrastive loss, NeurIPS 2021)?
+ Does ADDEDGE also lead to similar degenerate spectra? If yes, what can explain the difference in their performance?
+ The previous questions leads to a more important question. Existing works such as (Analyzing Data-Centric Properties for Graph Contrastive Learning, NeurIPS 2022) have argued that off-the-shelf augmentation strategies can lead to task-irrelevant invariances and sometimes even create erroneous signals (e.g., removing or adding an edge changing the molecule property). How does one ensure that these generic augmentations (e.g., DROPEDGE) are safely used in practice and more importantly, can spectral augmentations provide any complementary benefits. For example, can we use both edge perturbations and spectral augmentations simultaneously?

**Questions:**

Please see weaknesses.

---

### Official Review · Reviewer_W2wJ · 2024-11-07

**Soundness:** 2
**Presentation:** 3
**Contribution:** 3
**Rating:** 5
**Confidence:** 3

**Summary:**

This paper discusses an interesting problem in graph self-supervised learning, i.e., spectral augmentations are really necessary for contrast-based graph self-supervised learning. To fully evaluate the proposed question, the authors revisit simple edge perturbation for various graph learning tasks and conclude that these conventional approaches yield superior performance while demanding significantly fewer computational resources compared to existing spectral augmentation methods.

**Strengths:**

+ Overall the paper is written well and is intuitive. It is reasonable that the authors conduct comparisons among different graph related augmentations for graph self-supervised learning across various tasks.
+ The authors provide comprehensive comparisons not only from graph learning tasks but backbone models.

**Weaknesses:**

- Although the authors involve multiple datasets, I wonder if the authors can get similar conclusions over OGB datasets? For OGB datasets, they also have node-level, edge-level, and graph-level tasks.
- The baselines (i.e., the backbones) are not state-of-the-art. Can the authors evaluate the proposed idea on cutting-edge baselines as well? I think it will make the conclusion and evaluation more promising.
- Ablation studies should involve more datasets.
- Code is not available.

**Questions:**

Please see comments in the Weaknesses part.

---

### Official Review · Reviewer_YmCC · 2024-11-07

**Soundness:** 3
**Presentation:** 3
**Contribution:** 3
**Rating:** 6
**Confidence:** 2

**Summary:**

In this submission, the authors claim that spectral augmentations are unnecessary in contrast-based graph self-supervised learning.
To support this claim, the authors thoroughly study the relationship between spectral characteristics and the performances of current methods.
The authors also show the strategies of random edge dropping and random edge adding are simple yet effective.
The findings of this submission can be a good help to future graph SSL implementation.

**Strengths:**

1. The submission is well-motivated. It provides a detailed explanation of the discussions about the necessity of spectral augmentations in contrast-based graph SSL.
2. The authors conduct a comprehensive experimental analysis across multiple datasets to support their claim.

**Weaknesses:**

I’m a little bit concerned about the baselines for the spectral augmentation strategies. In line 316, the authors say “there are only three very relevant studies on spectral augmentation strategies of CG-SSL to the authors’ best knowledge, i.e., SPAN, SpCo and GASSER.” However, it seems there are other existing works[1-4] and the authors have cited some of them. Could the authors discuss why these works are “not that relevant” to spectral augmentation strategies of CG-SSL and why they didn’t consider them as baselines?

In Table 1, the definitions of $n$ and $m$ are missing.

[1] Yang, Kaiqi, Haoyu Han, Wei Jin, and Hui Liu. "Augment with care: Enhancing graph contrastive learning with selective spectrum perturbation." arXiv preprint arXiv:2310.13845 (2023).

[2] HaoChen, Jeff Z., Colin Wei, Adrien Gaidon, and Tengyu Ma. "Provable guarantees for self-supervised deep learning with spectral contrastive loss." Advances in Neural Information Processing Systems 34 (2021): 5000-5011.

[3] Shou, Yuntao, Xiangyong Cao, and Deyu Meng. "Spegcl: Self-supervised graph spectrum contrastive learning without positive samples." arXiv preprint arXiv:2410.10365 (2024).

[4] Zhang, Yifei, Hao Zhu, Zixing Song, Piotr Koniusz, and Irwin King. "Spectral feature augmentation for graph contrastive learning and beyond." In Proceedings of the AAAI Conference on Artificial Intelligence, vol. 37, no. 9, pp. 11289-11297. 2023.

**Questions:**

Please see above.

---

### Official Review · Reviewer_v2yb · 2024-11-10

**Soundness:** 2
**Presentation:** 3
**Contribution:** 2
**Rating:** 3
**Confidence:** 4

**Summary:**

This paper argues that simple edge-perturbation based methods are as effective, and sometimes more effective, than spectral augmentation-based methods for graph self-supervised learning. Based on these observations, the paper takes a strong position against the relevance of spectral augmentations-based self supervised learning.

**Strengths:**

This paper provides a comprehensive empirical comparison between spectral augmentation-based and edge perturbations-based graph self-supervised learning methodologies. Experiments demonstrate that simple edge perturbations perform as well or outperform more computationally complex spectral augmentations. The paper also brings into focus the shortcomings in theoretical understanding of spectral augmentation-based graph self-supervised learning.

**Weaknesses:**

1. If novelty of contributions to machine learning is the criterion for acceptance, the paper falls quite short on it. In my opinion, there is no sufficiently novel methodological or theoretical insight that could be leveraged by a researcher in this field.

2. Spectral augmentation-based graph self-supervised learning is effective in some experiments tabulated in Table 3. I appreciate the strong position taken by authors based on their experiments. However, spectral analysis of graph neural networks has enriched the theoretical understanding of GNN models from different perspectives. While bringing focus to the current shortcomings of spectral augmentations is relevant, the paper does not provide a conclusive (theoretical) evidence that would warrant a categorical dismissal of spectral augmentation-based graph self-supervised learning algorithms.

**Questions:**

See weaknesses.

---

### Note · Authors · 2024-11-22

I have read and agree with the venue's withdrawal policy on behalf of myself and my co-authors.